# The Rapid Activation of MYDGF Is Critical for Cell Survival in the Acute Phase of Retinal Regeneration in Fish

**DOI:** 10.3390/ijms26157251

**Published:** 2025-07-27

**Authors:** Kayo Sugitani, Yuya Omori, Takumi Mokuya, Serika Hosoi, Haruto Kobayashi, Koki Miyata, Yuhei Araiso, Yoshiki Koriyama

**Affiliations:** 1Department of Clinical Laboratory Science, Graduate School of Medical Science, Kanazawa University, 5-11-80 Kodatsuno, Kanazawa 920-0942, Ishikawa, Japan; yuya0505.medic@gmail.com (Y.O.); m.mtaku916@gmail.com (T.M.); hosoiserika@gmail.com (S.H.); halu86haru@gmail.com (H.K.); kokimiyata0916@gmail.com (K.M.); araiso@staff.kanazawa-u.ac.jp (Y.A.); 2Graduate School and Faculty of Pharmaceutical Sciences, Suzuka University of Medical Science, 3500-3 Minamitamagaki, Suzuka 513-8670, Mie, Japan; koriyama@suzuka-u.ac.jp

**Keywords:** MYDGF, HSF1, Bcl-2, caspase-3, TNFα, retina, optic nerve regeneration, zebrafish: cell survival, CRISPR/Cas9 system

## Abstract

Myeloid-derived growth factor (MYDGF), named in reference to its secretion from myeloid cells in bone marrow, is a novel protein with anti-apoptotic and tissue-repairing properties. MYDGF is found in various human tissues affected by different diseases. To date, however, MYDGF expression has yet to be reported in the nervous system. Herein, we demonstrate for the first time that MYDGF mRNA levels increased in the zebrafish retina 1 h after optic nerve injury (ONI). MYDGF-producing cells were located in the photoreceptors and infiltrating leukocytic cells. We prepared the retina for MYDGF gene knockdown by performing intraocular injections using either MYDGF-specific morpholino or the CRISPR/Cas9 system. Under these MYDGF-knockdown retinal conditions, anti-apoptotic Bcl-2 mRNA was suppressed; in comparison, apoptotic caspase-3 and inflammatory TNFα mRNA were significantly upregulated in the zebrafish retina after ONI compared to the control. Furthermore, heat shock factor 1 (HSF1) was evidently suppressed under these conditions, leading to a significant number of apoptotic neurons. These findings indicate that MYDGF is a key molecule in the stimulation of neuronal regeneration in the central nervous system.

## 1. Introduction

Myeloid-derived growth factor (MYDGF), also known as the open reading frame on chromosome 19 (C19orf10) in humans, is a novel protein secreted from myeloid cells in bone marrow stromal cells affected by different diseases [1,2,3,4]. MYDGF was first detected in the heart after myocardial infarction, where it was shown to protect the heart and enhance its repair; it is produced by bone marrow-derived monocytes and macrophages [1]. Its strong cardiomyocyte-protective and angiogenesis-promoting activities in myocardial infarction resulted in this protein being named myeloid-derived growth factor (MYDGF). Subsequently, MYDGF was identified in other tissues and linked to various human diseases, including heart failure [4,5,6], atherosclerosis [3,4], kidney dysfunction [4,7,8], inflammation [3,4,7,9], and cancer [4,10,11,12]. Excluding cancer, MYDGF secretion has been demonstrate to ameliorate these diseases through its anti-apoptotic and anti-inflammatory effects and tissue repair properties [1,4,5,7,13]. However, the deletion of MYDGF has been shown to exacerbate these diseases [4,8,14,15].

To the best of our knowledge, MYDGF secretion in the nervous system has yet to be reported. As previously established, mammalian central nervous system (CNS) neurons cannot survive and eventually die following nerve injury [16,17,18,19,20,21,22,23,24]. In contrast, fish CNS neurons can survive, their axons can regrow, and their CNS functions can be restored following nerve injury [25,26,27,28,29]. Using a zebrafish optic nerve injury (ONI) model, it has been established that anti-apoptotic effects occurring within the first 24 h post-injury play a critical role in ensuring the survival of neural cells [30,31]. Herein, we demonstrate significant MYDGF expression in the zebrafish retina 1 h after ONI. In the zebrafish retina, the results of MYDGF knockdown experiments confirmed that MYDGF plays a crucial role in inducing anti-apoptotic, tissue repair, and anti-inflammatory effects immediately after ONI. Its activation influenced the upregulation of heat shock factor 1 (HSF1), which was observed in the zebrafish retina between 0.5 and 24 h after ONI [30,31,32,33]. As a result, this interaction contributed to enhanced cell survival. The results of this study provide the first source of evidence demonstrating that MYDGF is rapidly expressed following nerve injury, contributing to the survival of neural cells and preventing apoptosis.

## 2. Results

### 2.1. Rapid Increase in Myeloid-Derived Growth Factor (MYDGF) Expression in the Zebrafish Retina Immediately After ONI

Changes in MYDGF gene expression in the retina following optic nerve injury (ONI) were analyzed through real-time PCR using gene-specific primers (see Table 1). MYDGF mRNA levels peaked at 1 h after its upregulation and decreased to control levels 12–24 h after ONI (Figure 1a). During this period, a second low peak was observed 6 h after ONI (Figure 1a). These MYDGF mRNAs were expressed in photoreceptors just below the pigment epithelium (PE) in outer nuclear layers (ONLs), the upper edge of outer plexiform layers (OPLs), and inner plexiform layers (IPLs) in the retina, as demonstrated through in situ hybridization (Figure 1b, arrowheads). However, no positive signals were observed for the control retina (0 h) or 1 h retinal tissue reaction when using sense probes. Immunohistochemical staining analysis results also confirmed that MYDGF protein expression after ONI changed similarly to the mRNA levels (Figure 1c). In the untreated control retina (0 h), no MYDGF signal was observed. However, 1 h after ONI, significant staining was observed in the photoreceptor layer in ONLs, the upper edge of OPLs, and IPLs. This increase in MYDGF expression was quite weak after 12 h of ONI (Figure 1c, 12 h) and almost disappeared after 24 h. Western blot analysis results similarly revealed that MYDGF expression peaked at 1 h after ONI and gradually declined thereafter, consistent with the immunohistochemistry results (Figure 1d).

### 2.2. MYDGF-Producing Cells in the Zebrafish Retina After ONI

To determine the exact location of MYDGF protein secretion in the zebrafish retina 1 h after ONI, we performed double immunofluorescence staining of MYDGF with DAPI, Lcp1, and other marker proteins. DAPI is a fluorescent dye used to stain cell nuclei, and Lcp1 is a leukocytic marker protein [34,35]. The cells positive for the MYDGF protein included (1) photoreceptors, (2) the border of the ONL and OPL, and (3) the cells in the IPL (Figure 2a).

Fluorescent double staining for the photoreceptor markers rhodopsin and MYDGF revealed that MYDGF signals were localized just beneath the rhodopsin-positive outer segment layer of the ONL (Figure 2b). This signal may originate from substances secreted by the strongly stained region along the boundary between the outer nuclear layer (ONL) and inner nuclear layer (INL). However, based on the presence of MYDGF mRNA revealed through in situ hybridization, as shown in Figure 1b, it is more likely that this signal is produced directly by the photoreceptors. Notably, MYDGF expression was not detected in the ganglion cell layer (GCL) at this time point.

The cells located at the border in the ONL and OPL and also the IPL were positively stained with Lcp1 and for MYDGF (Figure 2c). The border in the ONL and OPL was also stained with the Iba1 antibody (Figure 2d, arrowhead), which serves as a marker for microglia and macrophages [36,37,38]. Based on the results of fluorescent double staining for GFAP (glial fibrillary acidic protein) and MYDGF, it was concluded that MYDGF was not produced in Müller cells (Figure 2e). Following GFAP fluorescent staining, the ONL appeared red; however, similar staining was observed in the negative control using Alexa Fluor 594 without the primary antibody, as shown in Appendix A. This signal was thus considered to be nonspecific fluorescence. The pigment epithelium (PE) was also determined not to produce MYDGF based on the brightfield and MYDGF fluorescence staining observations using the same slides (Figure 2f).

### 2.3. The Effect of MYDGF-Specific Morpholino (MYDGF-MO) in the Zebrafish Retina After ONI

MYDGF knockdown experiments are often performed to investigate the roles of MYDGF in living cell systems [4,8,14,15]. In the present study, we prepared the retina in MYDGF-specific morpholino (MYDGF-MO). Intraocular injections of MYDGF-MO 20 h before ONI completely suppressed MYDGF mRNA expression (Figure 3a, MYDGF-MO).

However, strong activation of MYDGF was observed in the standard-MO (Std-MO) injection group 1 h after ONI (Figure 3a). The anti-apoptotic factor Bcl-2 was completely suppressed in the MYDGF-MO injection group (MYDGF-MO) compared to the control, i.e., the standard-MO group (Std-MO), 1 h after ONI (Figure 3b). MYDGF-MO remarkably activated the apoptosis-inducing factor caspase-3 in the zebrafish retina 1 h after ONI; in comparison, the standard-MO-injected control group exhibited steady MYDGF expression (Figure 3c). Similarly, the inflammatory cytokine tumor necrosis factor α (TNFα) was noticeably upregulated in the MYDGF-MO-injected group compared to the control group (Std-MO) 1 h after ONI (Figure 3d). Similar changes were observed at the protein level. Protein expression levels were analyzed using Western blotting, and the results are presented in Appendix A.

### 2.4. The Effect of MYDGF Knockdown (KD) Using the CRISPR/Cas9 System in the Zebrafish Retina After ONI

To further validate the MYDGF inhibition data obtained with MYDGF-MO in the retina after ONI, we performed MYDGF knockdown experiments using the CRISPR/Cas9 system. First, to determine whether MYDGF mRNA expression was suppressed, we injected a mixture of MYDGF-specific single guide RNAs (sgRNAs) and Cas9 nucleases into the eye 1 h before ONI (see Appendix A).

**Figure 4 ijms-26-07251-f004:**
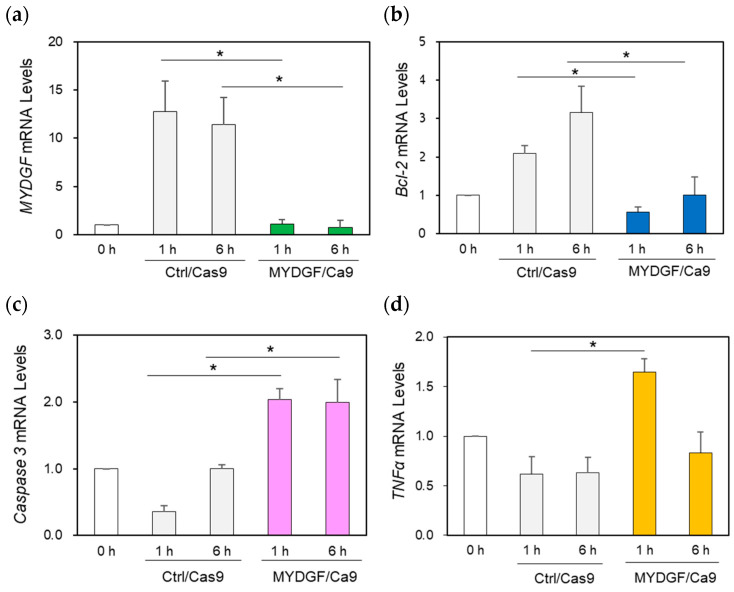
Effect of MYDGF knockdown with the CRISPR/Cas9 system using the intravitreal injection method. (**a**) Injection of MYDGF-specific sgRNA/Cas9 1 h before ONI markedly suppressed MYDGF mRNA expression (MYDGF/Cas9) 1 h and 6 h after ONI. (**b**) Under MYDGF inhibition conditions, Bcl-2 was also noticeably suppressed (MYDGF/Cas9) 1 h and 6 h after ONI. (**c**) Caspase-3 mRNA expression significantly increased at 1 h and 6 h in the MYDGF/Cas9 treatment group after ONI compared to the control group (Ctrl/Cas9). (**d**) The inflammatory factor TNFα was noticeably upregulated under the suppression of MYDGF expression 1 h after ONI (MYDGF/Cas9) compared to the control (Ctrl/Cas9). Statistical analysis was performed using one-way ANOVAs, followed by Scheffe’s multiple-comparison tests. Data are expressed as the means ± SEMs (*n* = 5–6), with statistical significance set at * *p* < 0.05.

In this experiment, electrophoresis was not performed; only injections were administered into the eye. Although the use of this method caused minimal damage to the eye and demonstrated lower overall effectiveness in inhibiting MYDGF compared to MO treatment, the proportion of individuals with inhibited MYDGF expression was similar. The analysis of MYDGF expression in the retinal samples 1 h and 6 h after ONI showed that MYDGF mRNA expression was significantly suppressed compared to that in the control/Cas9 (Ctrl/Cas9) injection group (Figure 4a). Under these conditions, we examined the expression of Bcl-2, caspase-3, and TNFα. Regarding the MYDGF/Cas9 injection group, the results presented in Figure 4b demonstrate that the anti-apoptotic factor Bcl-2 was noticeably suppressed in the retina 1 h and 6 h after ONI. The apoptotic factor caspase-3 was significantly upregulated at both 1 h and 6 h, and the inflammatory factor TNFα exhibited significant upregulation at 1 h after ONI compared to the control (Ctrl/Cas9) (Figure 4c,d). The efficiency of genome editing mediated by the CRISPR/Cas9 system is presented in Appendix A.

### 2.5. The Effect of Increasing the Number of Apoptotic Cells and Retinal Tissue Rupture in the MYDGF Knockdown Retina Produced via MYDGF-MO and the CRISPR/Cas9 System After ONI

After successfully producing retinal samples of MYDGF knockdown (KD) (Figure 3a and Figure 4a), we examined the damage to retinal cells and tissue under these conditions. 

**Figure 5 ijms-26-07251-f005:**
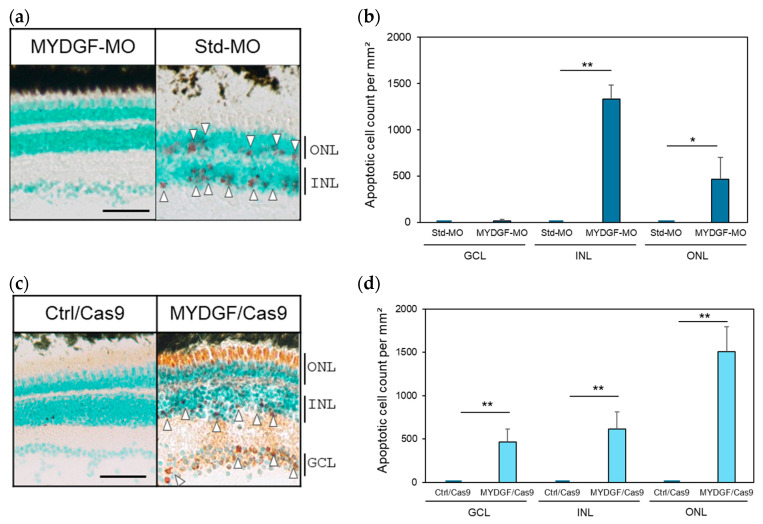
Suppression of MYDGF expression after ONI increased the number of apoptotic cells in the retina and disrupted the retinal layer structure. (**a**) Injection of MYDGF-specific MO (MYDGF-MO) significantly increased the number of red-stained apoptotic cells 3 h after ONI (white arrowhead); in comparison, no apoptotic cells were observed in the control-MO (Std-MO) injection group. (**b**) Apoptotic cells were quantified as the number of apoptotic-positive cells per mm^2^ in the retina. (**c**) Administration of MYDGF-specific sgRNA via the CRISPR/Cas9 system (MYDGF/Cas9) also significantly increased the number of red-stained apoptotic cells (white arrowhead), and most of the photoreceptors were stained red in the ONLs. No apoptotic cells were observed in the control/Cas9 injection group (Ctrl/Cas9). (**d**) Apoptotic cells in the retina were quantified as the number of apoptotic cells per mm^2^ following CRISPR/Cas9 system administration. Each nuclear layer of the retina was stained with 3% methyl green for counterstaining. Data are presented as the mean ± SEM (*n* = 5–6) of three independent experiments and were analyzed using one-way ANOVA. Statistical significance was set at ** *p* < 0.01 and * *p* < 0.05. ONL: outer nuclear layer; INL: inner nuclear layer; GCL: ganglion cell layer. Scale bar = 50 µm.

Regarding MYDGF-KD, a significant number of apoptotic cells were evaluated using the TUNEL method. TUNEL-positive cells, indicated by red signals marked with white arrowheads, were observed in the retinal cell layers after ONI in both the MYDGF-MO- and the CRISPR/Cas9 system-treated groups (Figure 5a,c). These damage signals were not observed in the retina treated with standard-MO (Std-MO) and control Cas9 (Ctrl/Cas9) injections. In the MYDGF-MO-treated group, few apoptotic cells were observed in the GCL, making it appear similar to the control group (Std-MO) at first glance (Figure 5b). This particular observation was attributed to severe tissue damage, resulting in the absence of the GCL and a markedly reduced number of retinal ganglion cells that could be quantified (Figure 5b).

### 2.6. The Effects of MYDGF Knockdown (KD) on Heat Shock Factor 1 (HSF1) Expression in the Zebrafish Retina After ONI

In our previous study, we demonstrated that heat shock factor 1 (HSF1) mRNA was upregulated in the zebrafish retina in the acute phase after ONI [30,31,32,33]. HSF1 mRNA activation was found to peak at 6 h after ONI [30,32]. Focusing on the different peak times of MYDGF and HSF1 activation, we examined the effects of MYDGF-KD on HSF1 mRNA expression in the zebrafish retina after ONI. MYDGF-MO completely suppressed the peak activation of HSF1 in the zebrafish retina 6 h after ONI (Figure 6a). Significant suppression of HSF1 expression was observed in the MYDGF-KD group, which was treated using the CRISPR/Cas9 system (Figure 6b).

## 3. Discussion

### 3.1. Rapid and Transient Activation of MYDGF Expression Plays a Critical Role in the Acute Phase of Retinal Regeneration in Zebrafish

MYDGF mRNA levels increased more than 10-fold at 1 h after ONI, returned to control levels by 3 h, increased again by 3–5-fold at 6 h, and subsequently declined to control levels within 12–24 h (Figure 1a). During this early stage (1 h after ONI), MYDGF-producing cells were limited to a subset of photoreceptors and infiltrating leukocytic cells in the ONL–OPL border and IPL (Figure 2); in comparison, no signal was detected in the ganglion cell layer (GCL). The inhibition of MYDGF expression using MYDGF-specific MO or the CRISPR/Cas9 system clearly demonstrated the anti-apoptotic and anti-inflammatory effects of MYDGF (Figure 3 and Figure 4). Due to the short duration of its activity and its limited coverage range, the anti-apoptotic effect of MYDGF could not be adequately explained. While MYDGF played a crucial role in promoting the survival of retinal neurons following ONI, its anti-apoptotic effects were not observed in the damaged ganglion cells post-ONI. These findings provide strong evidence that the rapid activation of MYDGF within 1 h is crucial for initiating the acute phase of retinal regeneration in zebrafish.

In this study, it was observed that MYDGF protein levels increased in parallel with mRNA induction following ONI. Although translation is generally expected to lag behind transcription, this rapid protein expression suggests a tightly coupled transcription–translation mechanism. Such synchrony is biologically plausible, particularly in the context of retinal regeneration, wherein immediate cellular responses may be required. Similar rapid translation has been reported for other stress-responsive factors, including HSF1 [39,40] and ATF4 (Activating Transcription Factor 4) [41], which are activated under various stress conditions such as oxidative stress and ER stress.

In our previous study, we identified two critical molecules specifically activated during the acute phase following ONI in the zebrafish retina: HSF1 and the Yamanaka factors [30]. In addition, we demonstrated that HSF1 directly activates all members of the Yamanaka factor family [30]. HSF1 is essential for cell survival [42,43,44]. Yamanaka factors, known for their role in cellular reprogramming [45,46], are crucial in facilitating the repair of injured retinal ganglion cells. The active time course of HSF1 demonstrated a 7–8-fold increase between 0.5 and 1 h, peaking with a 100-fold increase at 6 h, before returning to baseline levels within 48–72 h [30,32,33]. The temporal dynamics of HSF1 expression demonstrated that retinal ganglion cells and deep-layer cells in the INL were the first to express HSF1 at 0.5–1 h post-ONI [30]. Subsequently, photoreceptors located just beneath the pigment epithelium began expressing HSF1 at 2–3 h. By 6 h post-ONI, HSF1 expression was observed in all photoreceptors within the ONL and in all cells of the INL, indicating that the HSF1 protein affected all retinal cells [30]. Based on these observations, we propose that MYDGF and HSF1 work synergistically to exert anti-apoptotic effects and promote cell survival.

### 3.2. Acute Phase Dynamics of MYDGF and HSF1 Expression During Retinal Regeneration in Zebrafish

To validate the above hypothesis that MYDGF and HSF1 work synergistically to exert anti-apoptotic effects and promote cell survival, we conducted an experiment in which HSF1 expression was suppressed via intraocular injection of HSF1-MO. Of note, under the suppression of HSF1 mRNA expression, the expression of MYDGF was approximately 30-fold higher than that of the control (Appendix A). MYDGF peaked at 1 h post-injury (Figure 1a), preceding the >100-fold increase in HSF1 expression observed at 6 h post-injury [30]. This temporal sequence also supports the proposed upstream role of MYDGF in regulating HSF1 during the acute phase following ONI. Furthermore, MYDGF knockout resulted in a marked reduction in HSF1 expression (Figure 6a,b), further corroborating this hypothesis. The results of previous studies have demonstrated that HSF1 modulates a wide range of cellular pathways, including those involved in proteostasis, inflammation, and metabolic balance [47,48]. It is possible that MYDGF expression is upregulated to counteract the disruption of these pathways caused by HSF1 depletion (Appendix A). Once HSF1 expression reaches a certain level, a negative feedback mechanism is triggered, leading to a reduction in MYDGF expression. These results align with observations in zebrafish retinas following ONI, wherein MYDGF expression peaks at 1 h but rapidly declines thereafter, coinciding with the increased expression of HSF1 [30]. The activation of MYDGF in the retina, followed by the widespread and strong expression of HSF1, is considered a crucial event in protecting retinal neurons from stress and facilitating their survival after ONI. Future analyses focusing on MYDGF receptors and related mechanisms are expected to play a crucial role in advancing our understanding of its function. Although the specific receptor(s) for MYDGF remain unidentified, the results of recent structural analyses suggest a potential receptor-binding interface involving surface-exposed tyrosine residues and adjacent loop regions [2,4]. In other systems, MYDGF has been reported to activate intracellular signaling cascades such as PI3K-AKT and MAPK/STAT3 pathways, particularly in cardiac and endothelial cells [1,4]. These pathways are known to intersect with stress response mechanisms and may contribute to HSF1 activation in the retina. While speculative, these observations provide a basis for future investigation into the upstream signaling events linking MYDGF with HSF1 dynamics during retinal regeneration. MYDGF expression was elevated not only within 24 h after ONI but also in the following days, based on preliminary data. This study represents a pioneering effort in identifying the role of MYDGF in central nervous system tissue regeneration, opening new avenues for future research in this field.

## 4. Materials and Methods

### 4.1. Ethics Statement

All of the experimental procedures were approved by the Committee on Animal Experimentation of Kanazawa University, approval number #AP24-006. All animal care procedures were performed in accordance with the guidelines for animal experiments of Kanazawa University, and efforts were made to minimize suffering.

### 4.2. Animals

Adult zebrafish (Danio rerio; 3–4 cm in length) were used for all of the experiments. The zebrafish were anesthetized with 0.02% MS222 (Sigma-Aldrich, St. Louis, MO, USA) in 10 mM phosphate-buffered saline (PBS; pH 7.4), and their bilateral optic nerves were carefully crushed with forceps at 1 mm behind the eyeball. The zebrafish were then reared in 28 °C water until testing.

### 4.3. Tissue Preparation

Retinal tissue was prepared after optic nerve injury for histological analysis. Briefly, the eye was excised, and the lens was removed and fixed for 2 h at 4 °C in a 5% sucrose solution in 4% paraformaldehyde in PBS (pH 7.4). The tissue was then infused with increasing concentrations of sucrose (5% to 20%), followed by overnight incubation at 4 °C in 20% sucrose. Tissues were embedded in an optimal cutting temperature (OCT) compound (Sakura Fine Technical, Tokyo, Japan), and sections were prepared with thicknesses of 12–14 μm.

### 4.4. Total RNA Extraction and cDNA Synthesis

Zebrafish were anesthetized with a controlled overdose of MS222 (0.1%), and retinal tissue was removed at the appropriate time point. The removed retina was immediately homogenized in a solution of Isogen (Nippon Gene, Tokyo, Japan) using a BioMasher II (Nippi, Tokyo, Japan). We performed total RNA extraction according to the manufacturer’s instructions. Total RNA samples were subjected to first-strand cDNA synthesis using a Transcriptor High-Fidelity cDNA Synthesis Kit (Roche, Mannheim, Germany).

### 4.5. Quantitative Real-Time PCR

Quantitative real-time PCR was performed using a Power SYBR Green PCR Master Mix (Thermo Fisher Scientific, Waltham, MA, USA) and a Quant Studio 3 real-time system (Thermo Fisher Scientific). Gene-specific primers were designed using Primer 3 (version 2.6.1) and BLAST (version 2.14.0) based tools on the Zfin database to measure the expression levels of target genes in zebrafish. Expression levels were analyzed using the ΔΔCt method with glyceraldehyde 3-phosphate dehydrogenase (GAPDH) as the control gene. The accession numbers of the genes used in each experiment and the DNA sequences of the primer pairs are listed in Table 1.

### 4.6. Immunohistochemistry

Retinal sections were antigen-activated in 10 mM citrate buffer at 121 °C for 10 min. After washing and blocking, sections were incubated with primary antibodies overnight at 4 °C (anti-MYDGF, 1:1000, Atlas Antibodies, Stockholm, Sweden). Following incubation with a biotinylated secondary antibody (1:200, Vector Laboratories, Newark, CA, USA) for 2 h at room temperature, the bound antibodies were detected using horseradish peroxidase (HRP)-conjugated streptavidin (Nichirei Biosciences Inc., Tokyo, Japan) and 3-amino-9-ethyl carbazole (AEC; Nichirei Biosciences Inc., Tokyo, Japan).

The primary antibodies used for immunofluorescence staining included anti-MYDGF rabbit IgG (1:1000, Atlas Antibodies), anti-MYDGF rat IgG (1:250, BioLegend, San Diego, CA, USA), anti-rhodopsin mouse IgG (1:100, Abcam, Cambridge, UK), anti-GFAP mouse IgG (1:400, Merck, Darmstadt, Germany), anti-Iba1 rabbit IgG (1:1000, FUJIFILM, Osaka, Japan), and Anti-Lcp1 rabbit IgG (1:200, GeneTex, Alton Pkwy Irvine, CA, USA). The secondary antibodies used for fluorescence included Alexa Fluor 594 anti-rabbit IgG (1:2000), Alexa Fluor 488 anti-rabbit IgG (1:2000), Alexa Fluor 488 anti-rat IgG (1:2000), and Alexa Fluor 594 anti-mouse IgG (1:2000). All of the secondary antibodies were procured from Thermo Fisher Scientific (Waltham, MA, USA).

### 4.7. In Situ Hybridization

In situ hybridization was performed using a previously described method [49]. Tissue sections were rehydrated and treated with 5 mg/mL proteinase K (Invitrogen, CA, USA) at room temperature for 5 min. After acetylation and prehybridization, hybridization was performed with cRNA probes labeled with digoxigenin in a hybridization solution overnight at 42 °C. The following day, the sections were washed and treated with RNase A at 37 °C for 30 min. To detect the signals, the sections were incubated with an alkaline phosphatase-conjugated anti-digoxigenin antibody (Roche, Rot Kreuz, Switzerland) overnight at 4 °C and visualized with tetrazolium-bromo-4-chloro-3-indolylphosphate (Roche) as the substrate.

### 4.8. Intraocular Injection of MYDGF-Specific Morpholino (MYDGF-MO) into the Eyes

All of the vivo-morpholinos were purchased from GeneTools (Philomath, OR, USA). MYDGF-specific MOs were designed to inhibit the expression of the zebrafish MYDGF gene via the following sequence: 5′-GCCATGATGCTTTACAGACAGGAGA-3′ (MYDGF-MO). A standard vivo-MO (5′-CCTCTTACCTCAGTTACAATTTATA-3′) was used as the control (Std-MO). Using a Hamilton 33G neuron syringe, 0.5 µL volumes of all of the MO solutions (0.5 mM) were injected into the eyes 20 h before ONI. Retinal samples were removed at each time point, and total RNA was extracted.

### 4.9. MYDGF Gene Knockdown Experiment Using the CRISPR/Cas9 System

Four targets of the MYDGF gene were designed using CRISPRdirect software (https://crispr.dbcls.jp) and are presented in Appendix A. Template DNA was prepared from PCR products, which were purified using the PCR Clean-Up System (Promega, Madison, WI, USA), followed by the synthesis of four sgRNAs (single guide RNA, sgRNA1~sgRNA4) using the MEGA short script™ T7 transcription kit (Thermo Fisher Scientific, Waltham, MA, USA). Following purification with the Clean-Up Spin Column (mirVana miRNA Isolation Kit, Thermo Fisher Scientific, Waltham, MA, USA), the sgRNA concentrations were quantified. The efficiency with which the targeted DNA region was cleaved by each sgRNA/Cas9 was evaluated in vitro and confirmed using agarose gel electrophoresis (Appendix A). Subsequently, MYDGF knockdown efficiency was measured by injecting Cas9 nucleases (500 ng/µL, IDT, Coralville, IA, USA) and each sgRNA into the zebrafish eyes 1 h before the optic nerve was crushed. As sgRNA1 was the most efficient sgRNA (Appendix A), it was employed for all of the intraocular injections carried out using the CRISPR/Cas9 system [50,51].

### 4.10. Detection of Apoptotic Cells Through Terminal Transferase-Mediated dUTP Nick-End Labeling (TUNEL) Staining

We performed TUNEL staining to facilitate the detection of apoptotic cells in the retina after the MYDGF knockdown experiment using an In situ Apoptosis Detection Kit (TaKaRa, Shiga, Japan). Briefly, to remove endogenous peroxidase, retinal sections were treated with 0.3% H_2_O_2_ for 20 min after PBS washing. After treatment with a permeabilization buffer, the sections were reacted with a labeling mix solution containing terminal transferase and fluorescence dUTP for 75 min at 37 °C. For observation under visible light, the sections were reacted with biotinylated anti-FITC (1:100, Southern Biotech, Birmingham, AL, USA), HRP–streptavidin (Nichirei Bioscience, Tokyo, Japan), and AEC (3-amino-9-ethylcarbazole) substrate solutions (Nichirei Bioscience, Tokyo, Japan).

### 4.11. Western Blot Analysis

Retinal samples were immersed in lysis buffer (pH 7.5) [52] supplemented with a protease inhibitor cocktail (Nakarai Tesque, Kyoto, Japan), homogenized, and centrifuged at 15,000× *g* for 5 min. The resulting supernatant, containing 30 µg of protein, was mixed with Laemmli sample buffer and heat-treated at 95 °C for 5 min. The proteins were then separated via SDS-PAGE in a 10% gel and transferred onto a PVDF membrane (Bio-Rad, Hercules, CA, USA). The membrane was incubated in a blocking buffer (3% bovine serum albumin in PBS) for 1 h at room temperature and subsequently probed with a primary antibody diluted in blocking buffer overnight at 4 °C. The primary antibodies utilized included anti-MYDGF (as described above), anti-Bcl-2 (1:500, Santa Cruz Biotechnology, Dallas, TX, USA), anti-caspase 3 (1:500, Proteintech, Osemont, IL, USA), and anti-TNFα (1:500, Proteintech, Osemont, IL, USA). The membrane was additionally washed and incubated with an HRP-conjugated secondary antibody in blocking buffer for 1 h at room temperature. The signal was visualized using Clarity Western ECL Substrate with the ChemiDoc Imaging System (Bio-Rad, Hercules, CA, USA). Densitometric analysis of each protein band was performed using ImageJ software (version 1.54g; NIH, Bethesda, MD, USA). All of the experiments were conducted in triplicate or multiple replicates.

### 4.12. Indel Detection Assay

Indel mutations induced by CRISPR/Cas9 were assessed using the Rapid Indel Detection Kit (Nippon Gene, Tokyo, Japan) according to the manufacturer’s instructions. Genomic DNA was extracted from retinal tissue using the provided Template Prepper reagents, followed by PCR amplification of a 368 bp target region using a high-fidelity polymerase. The primers used for amplification were as follows: forward, 5′-AACTCCAGAACTGCCCCATC-3′; reverse, 5′-GCGAGTGCGGACACATAGAA-3′. The PCR products were subsequently denatured and re-annealed to form heteroduplexes, which were digested with T7 Endonuclease I. The resulting fragments were analyzed via agarose gel electrophoresis to detect indel formation. (Appendix A).

### 4.13. Statistical Analysis

The expression levels of MYDGF, Bcl-2, caspase-3, TNFα, and HSF1 are expressed as the means ± SEMs, and significant differences in mRNA expression were evaluated via one-way ANOVAs. A significance level of *p* < 0.05 was determined using IBM SPSS (29.0.2.0) Statistics software.

## 5. Conclusions

MYDGF significantly contributes to cell survival in the zebrafish retina through its rapid activation after optic nerve injury. The activation of MYDGF leads to the subsequent expression of HSF1, which may work synergistically with MYDGF to prevent cell death.

## Figures and Tables

**Figure 1 ijms-26-07251-f001:**
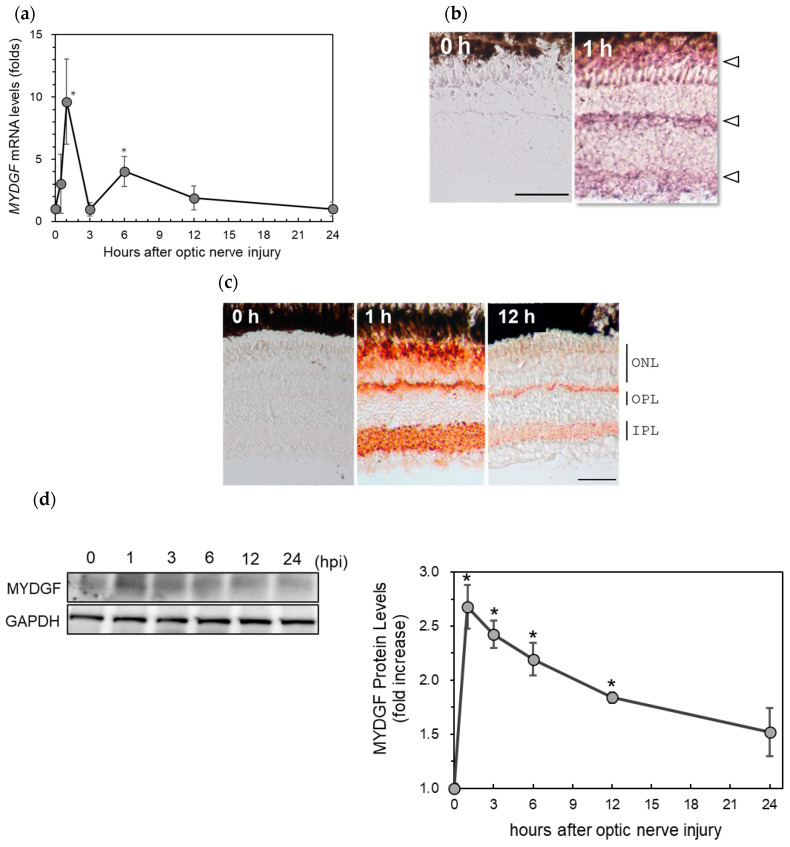
The upregulation of MYDGF (myeloid-derived growth factor) in the zebrafish retina after ONI (optic nerve injury). (**a**) MYDGF mRNA expression levels after ONI were determined using quantitative real-time PCR. Gene expression levels were normalized to GAPDH as an internal control, and fold changes were calculated relative to the 0 h (untreated) baseline group using the ΔΔCt method. Statistical analysis was performed using one-way ANOVAs, followed by Scheffe’s multiple-comparison tests. Data are expressed as the means ± SEMs (*n* = 6–7), with statistical significance set at * *p* < 0.05. (**b**) In situ hybridization of MYDGF in the zebrafish retina after ONI. MYDGF mRNA peaked in the retina 1 h after ONI, and it was observed in three locations in the retina, as indicated by the arrowheads. (**c**) Immunohistochemical staining of MYDGF in the zebrafish retina after ONI. Significant immunostaining peaked at 1 h in outer nuclear layers (ONLs), the border of the ONLs and outer plexiform layers (OPLs), and the inner plexiform layers (IPLs). Scale bar = 50 μm. Representative images from 3–4 independent experiments of in situ hybridization and IHC are shown. (**d**) Western blot analysis of MYDGF expression in the zebrafish retina at 0–24 h after ONI. MYDGF protein levels peaked in the retina 1 h after ONI. Data are expressed as the means ± SEMs (*n* = 3), with statistical significance set at * *p* < 0.05.

**Figure 2 ijms-26-07251-f002:**
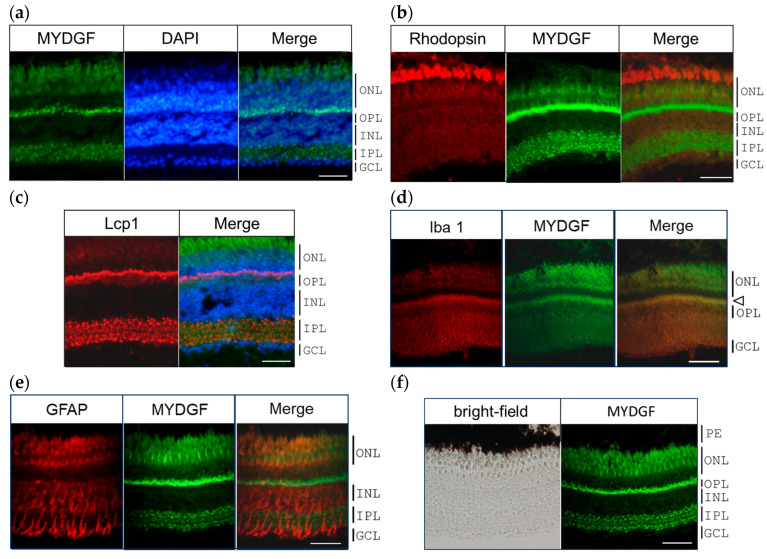
MYDGF-producing cells in the zebrafish retina 1 h after ONI. (**a**) Double fluorescence staining of MYDGF (green) and nuclear staining with DAPI (blue). The MYDGF-positive cells included photoreceptors in the ONL, the border of the ONL and OPL, and the cells in the IPL (Figure 2a). (**b**) Fluorescent double staining of the photoreceptor marker proteins rhodopsin (red) and MYDGF (green). (**c**) Triple fluorescence staining of MYDGF (green), nuclear staining with DAPI (blue), and leukocyte marker Lcp1 (red). Lcp1 clearly colocalized with the MYDGF-positive distribution at the ONL-OPL border regions and in the IPL. (**d**) The microglial marker protein Iba1 colocalized with the MYDGF-positive region at the ONL-OPL border regions (indicated by the arrowhead). (**e**) GFAP-positive Müller cells showed almost no co-localization with MYDGF. (**f**) The pigment epithelium (PE), observed as black under brightfield microscopy, was found to lack MYDGF production. Representative images from 3–4 independent experiments of IHC are shown. Scale bar = 50 μm.

**Figure 3 ijms-26-07251-f003:**
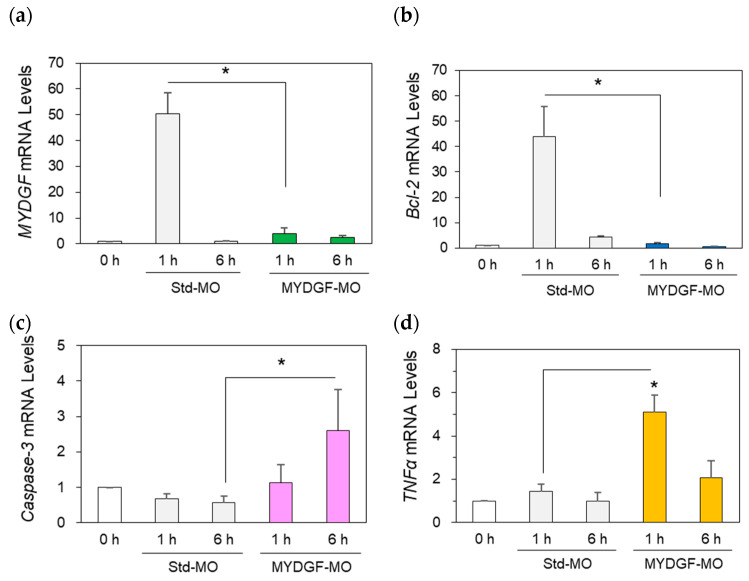
Inhibition of MYDGF expression induced neuronal apoptosis-inducing factors in the retina after ONI. Treatment with MYDGF-MO markedly suppressed MYDGF mRNA (**a**) and Bcl-2 mRNA (**b**) expression. Under MYDGF inhibition conditions, caspase-3 mRNA (**c**) and TNFα mRNA (**d**) expression noticeably increased 1 h and 6 h after ONI. Statistical analysis was performed using one-way ANOVAs, followed by Scheffe’s multiple-comparison tests. Data are expressed as the means ± SEMs (*n* = 6–7), with statistical significance set at * *p* < 0.05.

**Figure 6 ijms-26-07251-f006:**
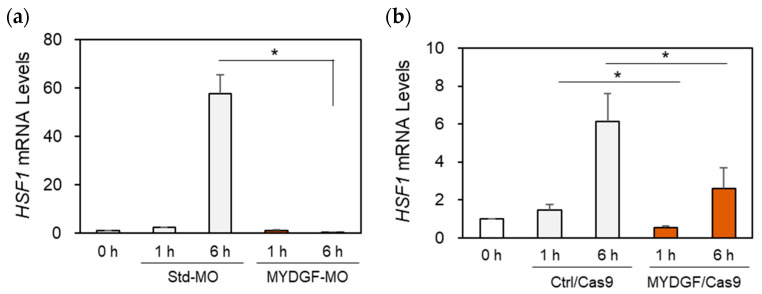
Effect of MYDGF knockdown on HSF1 expression after ONI. MYDGF-specific MO or MYDGF-specific sgRNA injections via the CRISPR/Cas9 system. (**a**) Knockdown with MYDGF-specific MO (MYDGF-MO) and (**b**) MYDGF-specific sgRNA injections using the CRISPR/Cas9 system (MYDGF/Cas9) markedly reduced HSF1 mRNA expression in the retina 6 h after ONI. Statistical analysis was performed using one-way ANOVAs, followed by Scheffe’s multiple-comparison tests. Data are expressed as the means ± SEMs (*n* = 5–6), with statistical significance set at * *p* < 0.05.

**Table 1 ijms-26-07251-t001:** Sequences of the PCR primers used in this study.

Gene	Accession No.	5′ Primer	3′ Primer	Purpose
*MYDGF*	NM_001002480	CAATTTTCTGCTCACGGTCA	CTTTGGCCACTGGCATCTAT	Real-time PCR
		CTGCTGCTGTTTGTTGTGCT	AGCAACATCCCTCTGTCCAC	In situ hybridization
*Bcl-2*	NM_001030253	CTGGAAAACTGGATCGAGGA	AAAACGGGTGGAACACAGAG	Real-time PCR
*Caspase-3*	NM_131877	TGCCAAGAAACAGATCCCCT	GCTGCTGACGTTCTCAAAGT	Real-time PCR
*TNFα*	NM_212859	AAAGTCGGGTGTATGGAGGG	TTGCCCTGGGTCTTATGGAG	Real-time PCR
*HSF1*	NM_001313736	GATCTGCTGGAGCCCAAA	TCGGCAGAACTTCTTTGGAA	Real-time PCR
*GAPDH*	NM_001115114.1	TCAGTCCACTCACACCAAGTG	CGACCGAATCCGTTAATACC	Real-time PCR

## Data Availability

https://kanazawa-u.repo.nii.ac.jp/?page=1&size=20&sort=custom_sort&search_type=0&q=0.

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
