# Peer review of "The Rapid Activation of MYDGF Is Critical for Cell Survival in the Acute Phase of Retinal Regeneration in Fish"

_ijms, 2025, doi:10.3390/ijms26157251_

Round 1

Reviewer 1 Report (Previous Reviewer 3)

Comments and Suggestions for Authors

This manuscript is interesting. Several comments could be addressed during revisions.

MYDGF mRNA levels are dynamically changed after optic nerve injury. 24 hours are considered long-term time point in this study. For the secondary effects by optic nerve injury, is it correct to set 24 hour as the final time point? It is also interesting to see 3 h after the injury levels are back to normal but 6 h there is some increase again.

mRNA time point and protein time point are exactly the same. Considering the translation process, it can be discussed.

In Figure 2, it is quite difficult to distinguish each labeling. Basically, whole layers are labeled regardless of the different antibody. It may need some negative control and positive control for each antibody in the supplementary file. Again, IBA1 and GFAP seem not clearly labelled. Cell morphology is not so clear. Rhodopsin should be with photoreceptor areas (outer retina only?).

1 h and 6 h timepoint has been selected. Is it related to the figure 1? Then how about 12 hour or 24 hour?

GAPDH is a correct control gene for qPCR in this case? It needs some data to show that GAPDH is suitable for one of the housekeeping genes in this model. Metabolic genes can be changed by optic nerve injury as far as the reviewer thinks.

Figure 4, TNF only has 1 hour data. This rationale is not so clear.

Efficiency of the gene modulation by CRISPR/CAS9 is not clearly presented in this study. Protein levels with retinal images could be supportive.

Author Response

Dear Reviewer,

Thank you very much for your editorial work on our manuscript titled “The Rapid Activation of MYDGF is Critical for Cell Survival in the Acute Phase of Retinal Regeneration in Fish” (Manuscript ID: ijms-3702271). 

We have carefully revised the manuscript in accordance with your valuable comments and suggestions.

1) MYDGF mRNA levels are dynamically changed after optic nerve injury. 24 hours are considered long-term time point in this study. For the secondary effects by optic nerve injury, is it correct to set 24 hour as the final time point? It is also interesting to see 3 h after the injury levels are back to normal but 6 h there is some increase again.

Fig. 1 Upregulation of MYDG mRNA in the zebrafish retina after optic nerve injury

(dpl; days post lesion)

→Thank you very much for your insightful comment regarding the temporal dynamics of MYDGF expression following optic nerve injury.

As you pointed out, the 24-hour time point may not fully capture the secondary effects of the injury. In fact, our preliminary experiments (Fig. 1) indicate that MYDGF expression continues to fluctuate beyond 24 hours. However, the physiological significance of these later changes remains unclear at this stage. Therefore, in the present study, we have focused on the early phase (within 24 hours) to provide a clearer picture of the immediate response.

We agree that the transient normalization at 3 hours followed by a secondary increase at 6 hours is intriguing, and it may reflect a complex regulatory mechanism. We plan to investigate the longer-term dynamics and potential physiological roles of MYDGF in future studies.

2) mRNA time point and protein time point are exactly the same. Considering the translation process, it can be discussed.

→We appreciate the reviewer’s observation regarding the timing of mRNA and protein level changes. While translation is generally expected to lag behind transcription due to intermediate processing steps, our data suggests that MYDGF exhibits a tightly linked transcription-translation relationship. This phenomenon is biologically plausible given MYDGF’s role in retinal regeneration, where rapid protein expression may be necessary for immediate cellular responses.

Several factors may contribute to this synchrony, including efficient translation machinery, enhanced mRNA stability, and post-transcriptional regulatory mechanisms. Similar rapid translation has been reported for other stress-responsive factors, such as HSF1 and ATF4, both of which are known to be promptly activated under cellular stress conditions. A note regarding this rapid translation has been added to the Discussion section 3.1.

3) In Figure 2, it is quite difficult to distinguish each labeling. Basically, whole layers are labeled regardless of the different antibody. It may need some negative control and positive control for each antibody in the supplementary file. Again, IBA1 and GFAP seem not clearly labelled. Cell morphology is not so clear. Rhodopsin should be with photoreceptor areas (outer retina only?).

→ Thank you for your valuable comments regarding Figure 2.

In response, we have repeated the immunostaining using a different rhodopsin antibody to improve specificity and localization to the photoreceptor layer. Additionally, the IBA1 staining has been re-performed to enhance clarity and cellular morphology. Regarding GFAP, we observed non-specific signals in the photoreceptor layer, which raised concerns about antibody specificity. Therefore, we conducted a negative control experiment using only the secondary antibody without the primary antibody, and have included the results in Supplementary Figure 1.

4) 1 h and 6 h timepoint has been selected. Is it related to the figure 1? Then how about 12 hour or 24 hour?

→ Thank you for your question regarding the selection of the 1-hour and 6-hour time points.

These time points were chosen based on our data shown in Figure 1, which demonstrates a significant increase in MYDGF expression at 1 h and 6 h following optic nerve injury. In contrast, at 12 h and 24 h, MYDGF expression levels had already declined to near control levels. Therefore, these later time points were considered less suitable for evaluating the suppressive effects on MYDGF expression, and were not included in the current analysis.

5) GAPDH is a correct control gene for qPCR in this case? It needs some data to show that GAPDH is suitable for one of the housekeeping genes in this model. Metabolic genes can be changed by optic nerve injury as far as the reviewer thinks.

→Thank you for raising this important point regarding the use of GAPDH as a housekeeping gene for qPCR normalization.

Based on our previous studies and the Western blot data shown in Figure 1, GAPDH has demonstrated stable expression in our optic nerve injury model. Although metabolic genes can be affected by injury, GAPDH has been widely validated in retinal and neural injury contexts (e.g., Casson et al., IOVS, 2004).

We agree that reference gene validation is essential, and we plan to assess additional candidates in future studies to further strengthen our normalization strategy.

6) Figure 4, TNF only has 1 hour data. This rationale is not so clear.

→Thank you for pointing out the limited time points shown for TNF in Figure 4.

In response, we have now included the 6-hour data. This time point was initially omitted because TNF expression did not show a statistically significant change compared to the control.

7) Efficiency of the gene modulation by CRISPR/CAS9 is not clearly presented in this study. Protein levels with retinal images could be supportive.

→ Thank you for your comment regarding the efficiency of gene modulation by CRISPR/Cas9.

To address this point, we have included additional data demonstrating MYDGF protein expression levels by Western blotting, as well as results from the indel detection assay. These findings are now presented in the supplementary figure 4, providing supportive evidence for successful gene disruption in the retina.

Reviewer 2 Report (New Reviewer)

Comments and Suggestions for Authors

In this manuscript, the function of MYDGF in retinal regeneration is explored in a zebrafish optic nerve injury (ONI) model. The authors provide strong evidence that MYDGF is rapidly upregulated following injury and participates in neuronal survival through the modulation of apoptotic and inflammatory pathways, possibly via HSF1 activity. The study is well designed, reports clear-cut data, and proposes a novel hypothesis that MYDGF plays a role in central nervous system repair processes. Nonetheless, some of these areas require more clarity, and others require a better mechanistic link and discussion.

1 Identification of MYDGF expression in the nervous system, retinal regeneration, specifically, is novel and can lead to new paths in neuroregenerative biology.

2. The contention that MYDGF generates the "explosive peak" of HSF1 is poorly substantiated. The paper lacks direct mechanistic information (e.g., receptor pathways, promoter activity) to justify this conclusion.

3.  Since it is a secreted factor, its receptor(s) or signaling intermediates need to be discussed even if speculatively. This is a major omission given the conclusions.

4. Changes in expression are presented in figures, but fold changes (e.g., for RT-qPCR) are hard to interpret without defined baselines. More clearly present normalization and control details in figure legends.

5. Since they confirm the qPCR results, a representative Western blot figure should be included in the main figures, not supplemental only.

6. The CRISPR/Cas9 method is reported to be less effective than MO, but efficiency was only assumed based on expression. Was indel formation or genome editing verified?

7. It is interesting but less well explored mechanistically that HSF1 knockdown increases the expression of MYDGF. The authors should discuss whether this is a feedback or compensatory mechanism and cite relevant literature.

8. The finding that GCL has a limited impact of MYDGF deserves more clear explanation in the results and discussion sections. Could this be due to timing, location, or limitations with the delivery approach?

9. Certain images, notably Figure 2 and Figure 5, need more annotation (i.e., cell layer identities) and greater resolution to assess colocalization completely.

10. The authors state use of ANOVA but do not always include the n values per group in each figure. This needs to be clearly stated.

11. The investigation is centered on the initial 24 hours following injury. Although this is justifiable, the authors ought to comment on whether more long-term outcomes were or could be assessed in future research.

12. "Rapid activation" is a frequently used term. It would be useful to define the precise temporal limits that this term encompasses in the context of zebrafish retinal response.

13. There is some repetitive phrasing ("significantly increased," "markedly suppressed"). Attempt to create more variety and flow in the Results and Discussion sections.

14. The manuscript lacks mention of mismatch-MO or CRISPR off-target analysis. This is important for validating specificity, especially in vivo.

15.  The authors allude to MYDGF's established functions in heart and kidney damage. It would make the manuscript stronger to briefly link this retinal research to potential translational avenues.

Comments on the Quality of English Language

The manuscript is generally well written and the language is clear and understandable. The authors convey their scientific findings with sufficient clarity and technical accuracy. However, there are several instances of awkward phrasing, redundancy, and overly repetitive sentence structures.

Author Response

Dear Reviewer,

Thank you very much for your editorial work on our manuscript titled “The Rapid Activation of MYDGF is Critical for Cell Survival in the Acute Phase of Retinal Regeneration in Fish” (Manuscript ID: ijms-3702271). 

We have carefully revised the manuscript in accordance with your valuable comments and suggestions.

  1. Identification of MYDGF expression in the nervous system, retinal regeneration, specifically, is novel and can lead to new paths in neuroregenerative biology.

→Thank you very much for your encouraging comment. We appreciate your recognition of the novelty and potential significance of MYDGF expression in the nervous system, particularly in the context of retinal regeneration. We believe that our findings may contribute to a deeper understanding of neuroregenerative mechanisms and open new avenues for therapeutic strategies targeting retinal and neural repair.

  1. The contention that MYDGF generates the "explosive peak" of HSF1 is poorly substantiated. The paper lacks direct mechanistic information (e.g., receptor pathways, promoter activity) to justify this conclusion.

→ Thank you for your insightful comment regarding the proposed relationship between MYDGF and the rapid activation of HSF1.

We acknowledge that the current study does not provide direct mechanistic evidence, such as receptor identification or promoter-level regulation, to fully substantiate this connection. To address this limitation, we have revised the title of Discussion section 3.2 to “Acute Phase Dynamics of MYDGF and HSF1 Expression During Retinal Regeneration in Zebrafish”, which more accurately reflects the descriptive nature of our findings. In future studies, we plan to investigate the underlying signaling pathways and transcriptional mechanisms to better define the link between MYDGF and HSF1 activation.

  1. Since it is a secreted factor, its receptor(s) or signaling intermediates need to be discussed even if speculatively. This is a major omission given the conclusions.

→Thank you for your insightful comment regarding the omission of receptor(s) or signaling intermediates associated with MYDGF. We agree that discussing these aspects, even speculatively, can strengthen the translational implications of our findings. In response, we have added the following discussion section to briefly consider potential downstream pathways and receptor candidates based on current literature.

 “Although the specific receptor(s) for MYDGF remain unidentified, recent structural analyses have suggested a potential receptor-binding interface involving surface-exposed tyrosine residues and adjacent loop regions [2]. In other systems, MYDGF has been reported to activate intracellular signaling cascades such as PI3K-AKT and MAPK/STAT3 pathways, particularly in cardiac and endothelial cells [1]. These path-ways are known to intersect with stress response mechanisms and may contribute to HSF1 activation in the retina. While speculative, these observations provide a basis for future investigation into the upstream signaling events linking MYDGF to HSF1 dynamics during retinal regeneration. “

  1. Changes in expression are presented in figures, but fold changes (e.g., for RT-qPCR) are hard to interpret without defined baselines. More clearly present normalization and control details in figure legends.

→Thank you for your valuable comment regarding the clarity of fold change interpretation in our qPCR data.

To address this concern, we have revised the figure legends to more clearly describe the normalization strategy and control conditions used. Specifically, gene expression levels were normalized to GAPDH as an internal control, and fold changes were calculated relative to the 0-hour (untreated) baseline group using the ΔΔCt method. These details have now been explicitly stated in the revised legends to improve transparency and interpretability.

  1. Since they confirm the qPCR results, a representative Western blot figure should be included in the main figures, not supplemental only.

→ Thank you for your suggestion regarding the presentation of Western blot data. In response to this, we have added the Western blot results to Figure 1(d).

  1. The CRISPR/Cas9 method is reported to be less effective than MO, but efficiency was only assumed based on expression. Was indel formation or genome editing verified?

→ To address this point, we have included additional data from the indel detection assay. These results are now presented in Supplementary Figure 4 and provide supportive evidence of successful gene disruption in the retina.

  1. It is interesting but less well explored mechanistically that HSF1 knockdown increases the expression of MYDGF. The authors should discuss whether this is a feedback or compensatory mechanism and cite relevant literature.

→ Thank you for your thoughtful comment regarding the observed increase in MYDGF expression following HSF1 knockdown.

We agree that this finding is intriguing and warrants further investigation. Although the current study does not provide direct mechanistic insights, the upregulation of MYDGF may reflect a compensatory or feedback response to the loss of HSF1-mediated stress regulation. Previous studies have shown that HSF1 modulates a wide range of cellular pathways, including those involved in proteostasis, inflammation, and metabolic balance (Zhang et al., 2025; Occhigrossi et al., 2021). It is possible that MYDGF expression is upregulated to counteract the disruption of these pathways caused by HSF1 depletion.

To address this point, we have added a speculative discussion in the revised manuscript and cited relevant literature to provide context for this potential regulatory relationship. We also plan to explore the transcriptional and signaling mechanisms underlying this response in future studies.

  1. The finding that GCL has a limited impact of MYDGF deserves more clear explanation in the results and discussion sections. Could this be due to timing, location, or limitations with the delivery approach?

→We appreciate the reviewer’s insightful observation regarding the limited impact of MYDGF in the GCL. As noted, our data show minimal MYDGF-related changes in the GCL within 24 hours following optic nerve injury. However, strong MYDGF protein expression was observed in the immediately adjacent inner plexiform layer (IPL) at this early time point. We speculate that this localized upregulation of MYDGF in the IPL may exert a protective effect on neighboring GCL neurons, potentially contributing to their survival. Supporting this, increased apoptotic cells were clearly observed in the GCL of MYDGF-MO and MYDGF/Cas9 treated groups (Figure 5), indicating that MYDGF activity may play a key role in suppressing apoptosis in this region. We will clarify these points in the revised Results and Discussion sections.

  1. Certain images, notably Figure 2 and Figure 5, need more annotation (i.e., cell layer identities) and greater resolution to assess colocalization completely.

→Thank you for your helpful comment regarding the clarity and annotation of Figures 2 and 5. In response, we have revised both figures to include additional annotations identifying key retinal cell layers to improve interpretability. Regarding Figure 2, we also repeated the experiment using an alternative anti-rhodopsin antibody with enhanced specificity. This change allowed us to obtain clearer staining images, which we believe facilitate better assessment of colocalization. We have updated the manuscript and figure panels accordingly.

  1. The authors state use of ANOVA but do not always include the n values per group in each figure. This needs to be clearly stated.

→Thank you for pointing out the importance of clearly stating the sample size per group in each figure. We agree that this information is essential for interpreting statistical analyses such as ANOVA. The sample sizes (n values) were originally included in the figure legends; however, we realize that they may not have been sufficiently emphasized or consistently presented across all figures. To address this, we have revised the figure legends to ensure that the n values per group are clearly and explicitly stated.

  1. The investigation is centered on the initial 24 hours following injury. Although this is justifiable, the authors ought to comment on whether more long-term outcomes were or could be assessed in future research.

→Thank you very much for your insightful comment regarding the temporal dynamics of MYDGF expression following optic nerve injury.

As you pointed out, it may not be possible to fully capture the secondary effects of damage at the 24-hour mark. In fact, preliminary experiments in this study have shown that MYDGF expression increases again several days after optic nerve injury. However, the physiological significance of these later changes remains unclear at this point. Therefore, in this study, we focused on the early stage (within 24 hours) in order to clearly understand the immediate response. I have added a brief note on this in the Discussion section.

  1. "Rapid activation" is a frequently used term. It would be useful to define the precise temporal limits that this term encompasses in the context of zebrafish retinal response.

→Thank you for your insightful comment. In our manuscript, the term “rapid activation” refers to molecular or cellular responses that occur within 1 h post-injury. We acknowledge that the temporal scope of this phrase could benefit from clarification, and we will revise the text in Discussion.

  1. There is some repetitive phrasing ("significantly increased," "markedly suppressed"). Attempt to create more variety and flow in the Results and Discussion sections.

→Thank you for your helpful suggestion. In response, we have revised the Results and Discussion sections to introduce greater variety in wording while maintaining scientific accuracy. We appreciate your insight, which has helped improve the clarity and flow of our manuscript.

  1. The manuscript lacks mention of mismatch-MO or CRISPR off-target analysis. This is important for validating specificity, especially in vivo.

→Thank you for your valuable comment regarding mismatch-MO and CRISPR off-target analysis. In our study, we employed standard control morpholinos (Std-MO), which are widely accepted as functional equivalents of mismatch-MOs for assessing non-specific effects in zebrafish. Additionally, we performed a T7 Endonuclease I (T7E1) assay to confirm indel formation at the intended target site. The results of this assay are presented in Supplementary Figure 4, demonstrating the specificity of our CRISPR-based genome editing. Specifically, indel detection analysis was conducted to evaluate the knockout efficiency of the Cas9 system: the control group yielded a single clear PCR band, whereas the MYDGF/Cas9-KO group exhibited two distinct bands, indicating indel mutations introduced at the target locus. We will revise the manuscript to clarify these points in the main text to ensure appropriate emphasis on experimental specificity.

  1. The authors allude to MYDGF's established functions in heart and kidney damage. It would make the manuscript stronger to briefly link this retinal research to potential translational avenues.

→Thank you for your thoughtful comment. We appreciate your suggestion to highlight the translational potential of MYDGF beyond the retina. As noted, MYDGF has been implicated in tissue repair in organs such as the heart and kidney. Given its involvement in stress response and regeneration, we agree that this retinal study may provide insights relevant to broader therapeutic applications. We will revise the Discussion section to briefly address possible translational avenues, including the potential relevance of MYDGF signaling in other contexts of tissue injury and recovery.

Round 2

Reviewer 2 Report (New Reviewer)

Comments and Suggestions for Authors

I have read the revised manuscript and the authors' comprehensive rebuttal letter. I thank the authors for their considerate and careful revisions. The authors have sufficiently responded to all issues identified in the original review, and the manuscript is indeed much better in scientific clarity and presentation.

Minor Comments:

1. Although most figures are well-labeled, please use consistent abbreviations throughout all figures (e.g., ONL, IPL, GCL) and include scale bars consistently and make them clear in all image panels.

2.  Throughout the main text, there are some Supplementary Figures (e.g., Suppl. Figure 5 demonstrating enhanced MYDGF following HSF1 knockdown) that would be improved by more overt cross-referencing and concise contextual interpretation to direct the reader.

3. In some places (e.g., line 348), terms such as "several-fold higher" may be substituted with specific fold-change values, if possible, to improve clarity and reproducibility.

Author Response

Dear Reviewer,

Thank you very much for your editorial work on our manuscript titled “The Rapid Activation of MYDGF is Critical for Cell Survival in the Acute Phase of Retinal Regeneration in Fish” (Manuscript ID: ijms-3702271). 

We have carefully revised the manuscript in accordance with your valuable comments and suggestions.

  1. Although most figures are well-labeled, please use consistent abbreviations throughout all figures (e.g., ONL, IPL, GCL) and include scale bars consistently and make them clear in all image panels.

→ Thank you for your valuable comments. We have carefully reviewed all figure panels and revised them accordingly. Consistent abbreviations (e.g., ONL, IPL, GCL) have been applied throughout, and scale bars have been added or adjusted to ensure clarity and uniform presentation.

  1. Throughout the main text, there are some Supplementary Figures (e.g., Suppl. Figure 5 demonstrating enhanced MYDGF following HSF1 knockdown) that would be improved by moreover cross-referencing and concise contextual interpretation to direct the reader.

→ Thank you for this helpful comment. In response, we have revised the main text to improve contextual interpretation and cross-referencing of Supplementary Figures. Specifically, we have inserted the sentence at line 348-352 to better guide the reader and clarify the relevance of Suppl. Figure 5.

  1. In some places (e.g., line 348), terms such as "several-fold higher" may be substituted with specific fold-change values, if possible, to improve clarity and reproducibility.

→Thank you for your suggestion. In response, we have replaced the phrase “several-fold higher” in line 348 with a more specific value: “approximately 30-fold higher,” to improve clarity and reproducibility as recommended.

This manuscript is a resubmission of an earlier submission. The following is a list of the peer review reports and author responses from that submission.

Round 1

Reviewer 1 Report

Comments and Suggestions for Authors

In this revised manuscript, entitled “The Rapid Activation of MYDGF is Critical for Cell Survival in the Acute Phase of Retinal Regeneration in Fish”, the authors analyzed the expression of myeloid-derived growth factor (MYDGF) following optic nerve injury (ONI) in the zebrafish retina. The results demonstrated that MYDGF is rapidly expressed in photoreceptors and infiltrating leukocytes as early as 1 hour post-ONI. Inhibition of MYDGF expression using morpholinos or CRISPR/Cas9 knockdown (KD) led to the downregulation of anti-apoptotic Bcl-2 mRNA, while Caspase-3 and TNF-alpha were upregulated. Additionally, MYDGF inhibition suppressed HSF1 expression, resulting in an increased number of apoptotic cells.

Major Comments:

Although this study is interesting and may be significant to the field of retinal regeneration, it lacks substantial evidence to support any potential underlying mechanisms.

1.- While the paper appears solid at this stage, it requires significant substantial editing for grammar, English usage, and overall readability.

2.-Figure 1c: The authors need to provide evidence for the identity and distribution of MYDGF-expressing cells throughout the visual system using immunofluorescence staining with appropriate cell markers and corresponding quantifications (e.g., violin plots). Additionally, do the retinal pigment epithelium (RPE) and Müller glia also express MYDGF?

3.-Figures 2a and 2b: What is the rationale for using Lcp1 instead of other markers? Also, DAPI should not be referred to as part of “triple immunofluorescence staining” (Line 85).

4.-Figure 3. Provide representative western blots and quantification analyses for each protein analyzed.

4.-Figure 5: Quantify the number of apoptotic cells for each condition.

5.-Supplementary Figure 1a: Consider moving this figure to the main Figure 1 for direct comparison.

6.-How was the ONI performed in mouse? It’s not described in the methods section.

Minor Comments:

Line 30: Add “in humans” after (C19orf10).

Introduction: The introduction should highlight the main conclusions, not just describe the results. What makes this study significant?

Grammar issue (Line 250): “In the cells produced by HSF1 mRNA” should be corrected to “In the cells expressing HSF1 mRNA.”

Line 257: How was stress measured? Please rephrase for clarity.

Line 82: The titles in the results section should highlight the major findings rather than merely describing the experiments.

Line 17 in abstract: Morpholinos do not cause deletion of the gene, they block the translation of mRNA, and downregulate target genes, please rephrase.

Comments on the Quality of English Language

While the paper appears solid at this stage, it requires significant substantial editing for grammar, English usage, and overall readability.

Author Response

Dear Reviewer,

Thank you very much for your editorial work on our manuscript titled “The Rapid Activation of MYDGF is Critical for Cell Survival in the Acute Phase of Retinal Regeneration in Fish” (Manuscript ID: ijms-3480852). 

We have carefully revised the manuscript in accordance with your valuable comments and suggestions

Major Comments:

1.- While the paper appears solid at this stage, it requires significant substantial editing for grammar, English usage, and overall readability.

-> Thank you for your comments regarding the grammar, English usage, and overall readability of the manuscript. Following your suggestions, I have thoroughly revised the paper, focusing on substantial English editing to improve its clarity, coherence, and overall flow.

2.-Figure 1c: The authors need to provide evidence for the identity and distribution of MYDGF-expressing cells throughout the visual system using immunofluorescence staining with appropriate cell markers and corresponding quantifications (e.g., violin plots). Additionally, do the retinal pigment epithelium (RPE) and Müller glia also express MYDGF?

-> Double fluorescent staining of retinal cells for marker proteins and MYDGF was conducted, and Figure 2 was revised to facilitate the identification of MYDGF-producing cells. The results of these experiments further suggest that pigment epithelium and Müller cells do not produce MYDGF.

3.-Figures 2a and 2b: What is the rationale for using Lcp1 instead of other markers? Also, DAPI should not be referred to as part of “triple immunofluorescence staining” (Line 85).

->MYDGF has been identified as a protein secreted by macrophages and monocytes during myocardial infarction, and it has been reported to protect myocardial cells from apoptosis. Therefore, in this study, we used Lcp1, a marker of leukocytes, to investigate the origin of this protein, given that the presence of MYDGF was confirmed in damaged retinal tissue.

To better reveal the layered structure of the retina, DAPI staining was performed. The results in Figure 2a indicate that the line-like signal observed between the ONL and OPL, where the strongest MYDGF-positive fluorescence is present, merges with DAPI-positive cells. This suggests that the signal is most likely secreted by individual accumulated leukocytes.

4.-Figure 3. Provide representative western blots and quantification analyses for each protein analyzed.

->Western blotting was performed to analyze changes at the protein level, and the results are presented in Supplementary Figure 2. These findings were consistent with the results obtained at the mRNA level.

4.-Figure 5: Quantify the number of apoptotic cells for each condition.

-> Figure 5 compares the presence of apoptotic cells under MYDGF expression suppression with those observed in the standard MO-treated group and the CRISPR/Cas9 control group. In both control groups, apoptotic cells were absent. Consequently, the comparison with the experimental group focused on qualitative observations of apoptotic cell presence, as quantitative comparisons were not feasible between the control and experimental groups.

5.-Supplementary Figure 1a: Consider moving this figure to the main Figure 1 for direct comparison.

->Following your suggestion, we have moved Supplementary Figure 1a to Figure 1d. This facilitates the contrast between the results of MYDGF immunostaining in zebrafish and mice.

6.-How was the ONI performed in mouse? It’s not described in the methods section.

->The method for creating a mouse optic nerve injury model has been newly described in the 'Materials and Methods' section 4.2, 'Animals.

Minor Comments:

Line 30: Add “in humans” after (C19orf10).

-> Based on your suggestion, we added “in humans” after (C19orf10) as follows,

“Myeloid-derived growth factor (MYDGF), also known as the open reading frame on chromosome 19 (C19orf10) in humans, is a novel---”

Introduction: The introduction should highlight the main conclusions, not just describe the results. What makes this study significant?

->The following text was added at the end of the introduction.

“This study is the first to demonstrate that MYDGF is expressed immediately upon nerve injury, contributing to the survival of neuronal cells and preventing them from undergoing apoptosis.”

Grammar issue (Line 250): “In the cells produced by HSF1 mRNA” should be corrected to “In the cells expressing HSF1 mRNA.”

->We corrected as follows,

In the cells expressing HSF1 mRNA, which were shown via in situ hybridization, ----

Line 257: How was stress measured? Please rephrase for clarity.

-> We corrected as follows,

“----- for cell survival against ONI stress.”  à    “----- for cell survival against ONI”

Line 82: The titles in the results section should highlight the major findings rather than merely describing the experiments.

->We corrected as follows,

“2.2 MYDGF-producing cells in zebrafish retina after ONI”

Line 17 in abstract: Morpholinos do not cause deletion of the gene, they block the translation of mRNA, and downregulate target genes, please rephrase.

->We corrected as follows,

“MYDGF gene knockdown by performing intraocular injections using either MYDGF-specific morpholino or the CRISPR/Cas9 system."

Reviewer 2 Report

Comments and Suggestions for Authors
  1. The introduction section lacks sufficient background information.All preliminary information is taken for granted, which makes it impossible for a reader, not particularly expert in the field, to follow and appreciate the work. It is recommended to supplement the research background with recent advances in retinal optic nerve injury and regeneration that are relevant to the study content.
  2. Line 20: The expression "1-6h" is inaccurate and should be "1h and 6h". Please check and correct this throughout the entire manuscript.
  3. Line 71-72: English abbreviations should only be defined with their full names when they first appear in the manuscript. Please conduct a full-manuscript check and make necessary corrections.
  4. Line 74-75: “Data are expressed as the mean ± SEM, with statistical significance set at *p < 0.05.”The figure legend mentions that significance analysis was performed, but no significance markers are shown in the Figure  Please correct.
  5. There is an error in line 100. Please identify and correct this mistake.
  6. Line 126-130: The figure legend contains excessive result descriptions. I suggest moving most of these descriptions to the Results section rather than keeping them in the figure legend.
  7. In Figure 4b, there appears to be a significant difference between theMYDGF/Cas9 injection group and control/Cas9 group at 1 hour after ONI, but the authors did not describe or analyze this potential significance. Please verify the accuracy of the raw data.
  8. Figure 6b has the same problem as Figure 4b, please make sure that all the data has been analyzed correctly.
  9. The discussion about MYDGF expressionin mouse retina (lines 236-239) should be presented in the Results section rather than the Discussion. In addition, the Discussion section lacks in-depth and broad discussion, instead containing too many result descriptions.
Comments on the Quality of English Language

The not exactly linear English form completes the difficulty.

Author Response

Dear Reviewer,

Thank you very much for your editorial work on our manuscript entitled “The Rapid Activation of MYDGF is Critical for Cell Survival in the Acute Phase of Retinal Regeneration in Fish” (Manuscript ID: ijms-3480852). 

We have made revisions based on your valuable comments and suggestions. Below, you will find a detailed response to each of your specific suggestions.

  1. The introduction section lacks sufficient background information. All preliminary information is taken for granted, which makes it impossible for a reader, not particularly expert in the field, to follow and appreciate the work. It is recommended to supplement the research background with recent advances in retinal optic nerve injury and regeneration that are relevant to the study content.

--> We have modified the introduction according to your suggestion.

  1. Line 20: The expression "1-6h" is inaccurate and should be "1h and 6h". Please check and correct this throughout the entire manuscript.

--> All instances have been corrected to "1 h and 6 h or deleted."

  1. Line 71-72: English abbreviations should only be defined with their full names when they first appear in the manuscript. Please conduct a full-manuscript check and make necessary corrections.

-->According to the "Instructions for Authors" in this journal, abbreviations should be defined the first time they appear in each of three sections: the abstract, the main text, and the first figure or table. Therefore, we also defined the abbreviation in the legend for Figure 1, lines 71-72.

  1. Line 74-75: “Data are expressed as the mean ± SEM, with statistical significance set at *p < 0.05.” The figure legend mentions that significance analysis was performed, but no significance markers are shown in the Figure 1.Please correct.

-->Due to an error, the mark was dropped. The figure1 has now been corrected.

  1. There is an error in line 100. Please identify and correct this mistake.

-->The extra text on line 100 has been erased.

  1. Line 126-130: The figure legend contains excessive result descriptions. I suggest moving most of these descriptions to the Results section rather than keeping them in the figure legend.

-->We have revised the manuscript by moving most of the descriptions from the figure legend to the Results section, as per the reviewer's suggestion.

  1. In Figure 4b, there appears to be a significant difference between theMYDGF/Cas9 injection group and control/Cas9 group at 1 hour after ONI, but the authors did not describe or analyze this potential significance. Please verify the accuracy of the raw data.

-->Thank you for your insightful observation regarding Figure 4b. In response to your comment, we have carefully reviewed the raw data to ensure its accuracy. Additionally, we have updated the Results section to include an analysis of the significance between the MYDGF/Cas9 injection group and the control/Cas9 group at 1 hour after ONI.

  1. Figure 6b has the same problem as Figure 4b, please make sure that all the data has been analyzed correctly.

-->Thank you for your valuable input on Figure 6b. In response to your comments, we thoroughly examined the raw data to confirm its accuracy. The Results section has been updated to provide an analysis of the statistical significance between the MYDGF/Cas9 injection group and the control/Cas9 group at 1 h and 6 h post-ONI.

  1. The discussion about MYDGF expression in mouse retina (lines 236-239) should be presented in the Results section rather than the Discussion. In addition, the Discussion section lacks in-depth and broad discussion, instead containing too many result descriptions.

-->The results of the mouse retina experiment have been relocated to Figure 1d for better clarity and presentation. Additionally, the Discussion section has been thoroughly revised to enhance the overall interpretation of the findings.

Reviewer 3 Report

Comments and Suggestions for Authors The manuscript "The Rapid Activation of MYDGF is Critical for Cell Survival in the Acute Phase of Retinal Regeneration in Fish" is interesting. Its observation is critical for understanding retinal regeneration.   However, the authors highly depend on the technique to check mRNA levels. Protein levels should be examined. Immunohistochemical analyses have limitations for quantity screening.      Housekeeping gene selection might need rationale.   Retinal vessels are not deeply considered, hence, experiments are recommended to screen retinal vessel changes too (optionally).     Data' each n number should be presented.   The CRISPR/Cas9 system's efficacy and efficiency have not been presented in the current study. Its validation should be included.    

Author Response

Dear Reviewer,

Thank you very much for your editorial work on our manuscript entitled “The Rapid Activation of MYDGF is Critical for Cell Survival in the Acute Phase of Retinal Regeneration in Fish” (Manuscript ID: ijms-3480852). 

We have made revisions based on your valuable comments and suggestions. Below, you will find a detailed response to each of your specific suggestions.

The manuscript "The Rapid Activation of MYDGF is Critical for Cell Survival in the Acute Phase of Retinal Regeneration in Fish" is interesting. Its observation is critical for understanding retinal regeneration. However, the authors highly depend on the technique to check mRNA levels. Protein levels should be examined. Immunohistochemical analyses have limitations for quantity screening. Housekeeping gene selection might need rationale.

-->Western blotting was performed to analyze changes at the protein level, and the results are presented in Supplementary Figure 2. These findings were consistent with the results obtained at the mRNA level.

Retinal vessels are not deeply considered, hence, experiments are recommended to screen retinal vessel changes too (optionally).

--> We acknowledge the reviewer's comment regarding retinal vessels. Although retinal vessels were not considered in this study, we plan to focus on this aspect in future research.

 Data' each number should be presented. The CRISPR/Cas9 system's efficacy and efficiency have not been presented in the current study. Its validation should be included.

-->Thank you for your valuable feedback. In response to your comments, I have added sentences discussing the efficacy and efficiency of the CRISPR/Cas9 system to the Results section. Injections into the eyeball may occasionally cause inflammation; however, positive effects were observed in 70–80% of the injection group. That said, the suppression effects of the CRISPR/Cas9 system were relatively lower when compared to specific MO injections.

Round 2

Reviewer 1 Report

Comments and Suggestions for Authors

No further comments or suggestions 

Author Response

Dear Reviewer,

We sincerely appreciate your time and effort in reviewing our manuscript.

Your valuable feedback and constructive comments have been instrumental in enhancing the clarity and quality of our work, titled “The Rapid Activation of MYDGF is Critical for Cell Survival in the Acute Phase of Retinal Regeneration in Fish” (Manuscript ID: ijms-3480852).

Once again, thank you for your thoughtful contributions to this process.

Best regards,

Kayo Sugitani, Ph.D.

Department of Clinical Laboratory Science

Kanazawa University Graduate School of Medical Science

Kodatsuno 5-11-80, Kanazawa, 920-0942, Japan

Tel: +81-76-265-2599    Fax: +81-76-234-4369
